# Key factors in supporting adolescents to achieve high self-esteem and a positive body image: A qualitative community-based study

Eva Artigues-Barberà[1,2,3], Glòria Tort-Nasarre[4,5,6], Mercè Pollina-Pocallet[3,7], Yvonne Ferrer Suquet[2,8], Aleix Ayats Pallés[3,7], Olívia Guasch Niubó[3,7], Marta Ortega-Bravo [3,9,10], Ester García-Martínez [1,3,11]*

1 Department of Nursing and Physiotherapy, Faculty of Nursing and Physiotherapy, University of Lleida, Lleida, Spain, 2 Atenció Primària, Gerència Territorial de Lleida, Institut Català de la Salut, Lleida, Spain, 3 Fundació Institut Universitari per a la recerca a l'Atenció Primària de Salut Jordi Gol i Gurina (IDIAPJGol), Barcelona, Spain, 4 Department of Nursing and Physiotherapy. Faculty of Nursing and Physiotherapy, University of Lleida, Barcelona, Spain, 5 SAP ANOIA, Gerència Territorial Catalunya Central, Institut Català de la Salut, Barcelona, Spain, 6 AFIN, Research Group and Outreach Centre, Universitat Autònoma de Barcelona, Barcelona, Spain, 7 Centre d'Atenció Primària Bellpuig, Institut Català de la Salut, Lleida, Spain, 8 Research Support Unit Lleida, Fundació Institut Universitari per a la recerca a l'Atenció Primària de Salut Jordi Gol i Gurina (IDIAPJGol), Lleida, Spain, 9 Centre d'Atenció Primària Almacelles. Institut Català de la Salut, Lleida, Spain, 10 Department of Medicine. Faculty of Medicine, University of Lleida, Lleida, Spain, 11 Health Care Research Group, GRECS, Biomedical Research Institute of Lleida IRB-Lleida, Lleida, Spain

* ester.garcia@udl.cat

## Abstract

### Introduction

Body dissatisfaction can be affect adolescents' mental health, leading to issues with body weight control, low self-esteem, and poor school performance.

### Methods

A total of 24 interviews with adolescents, parents, and teachers in Spain were conducted for this study. The study aimed to explore and compare the views of adolescents, parents, and teachers regarding interventions to improve self-esteem and positive body image in school-aged adolescents. A generic qualitative study design with purposive sampling was used, and the data was thematically analyzed.

### Results

Key barriers identified included lack of family communication, unfavourable family environments, and insufficient training. Social media and gender stereotypes, along with social rejection, were major challenges for adolescents. Facilitators included family involvement and using role models as a strategy. Proposed interventions focused on individual and group recreational activities starting at an early age, engaging adolescents in activities of interest, and promoting collaboration with families.

**Data availability statement:** All relevant data are within the paper and its Supporting Information files.

**Funding:** PICARD award of Provincial Council of Lleida (code 4R23/360). Initials of the author who received the award: MPP Grant number awarded: only 1 grant awarded to MPP The full name of the funder: Provincial Council of Lleida URL of the funder's website: https://www.diputaciolleida.cat/ Regarding the role of the funders, we confirm that the funders only provided financial support for publication and did not influence the study design, data collection and analysis, decision to publish, or preparation of the manuscript.

**Competing interests:** The authors have declared that no competing interests exist.

**Abbreviations:** BD, body dissatisfaction; BI, body image; BMI, body mass index; CSE, Compulsory Secondary Education; ED, eating disorders; PBI, positive body image; SE, self-esteem; SM, social media.

## Conclusion

This study identified barriers and facilitators proposing multi-level interventions that engage adolescents, families, educational institutions, and the community. Strengthening facilitators and reducing barriers should guide future public health policies.

## Public contribution

These findings may be useful for developing multilevel interventions aimed at improving body image and self-esteem, which could, in turn, prevent and reduce the severity of clinical disorders during adolescence.

## 1. Introduction

Adolescence is characterised by several physical, psychological, and social changes that affect young people's self-esteem (SE), making them vulnerable to sociocultural influences [1]. During adolescence, people live under continuous social pressure due to an ideal body beauty standard that is upheld – characterised by, for example, a slim and/or muscular body – which can influence self-perception and body image (BI) [1,2].

Body dissatisfaction (BD), which refers to negative perceptions and feelings about one's own body, is common among the adolescent population and is associated with low SE and the development of serious health problems such as unhealthy weight control behaviours [1], eating disorders (ED) [1], and psychological distress [3,4]. BI refers to the subjective perception individuals have of their own body. It is a complex construct that encompasses thoughts, feelings, evaluations, and behaviours related to the body [5]. BD largely stems from the discrepancy between the perception of one's own BI and an idealised image [6]. Rates of BD range from 30% to 75%, depending on age and gender, with higher prevalence during adolescence, early adulthood, and among women [7]. Factors contributing to BD include body mass index (BMI), family influences, media exposure, particularly SM, and SE.

In this regard, BD is negatively related to SE. SE is one of the most important self-referential variables in adolescence [8], defined as the concept an individual has of themselves and their abilities and personal qualities [9]. It is considered a subjective construct and does not necessarily reflect a person's objective characteristics or competencies, nor how they are evaluated by others [9]. Furthermore, it influences personal development and enables individuals to take actions, whether successful or not, within their immediate context [10]. Although some studies have not been able to clearly establish the cause-effect direction, or it remains controversial, SE has been linked to variables such as age [9], gender and sex [8,11,12], having homosexual relationships or being a virgin [11], intuitive eating [13], BMI [14], culture [15], empathy [8], life satisfaction [8], emotional regulation [16], emotional intelligence [17], self-concept [18], academic motivation [19], academic engagement [20], academic performance [8,21], SM use [22,23], social support [24], family and peer attachment [25], childhood abuse [11], mental health, symptoms of anxiety and depression [11], BI, and positive body image (PBI) [26].

PBI is defined as the love, respect, acceptance, and appreciation of one's own body, along with comfort in one's body regardless of its actual physical appearance, and the ability to interpret messages in a body-protective way [27]. In this way, a PBI becomes a protective factor for both physical health and psychological well-being, promoting body acceptance while acknowledging that negative thoughts about the body may arise [13,27]. Moving towards a PBI in adolescents is feasible and worthwhile, according to established evidence.

In Spain, since the onset of the COVID-19 pandemic in 2020, higher prevalence rates of anxiety, depressive symptoms, ED, self-harm, and suicidal behaviours have been noted among

adolescents, which, on many occasions, are a consequence of BD and low SE [28]. In this context, previous studies have shown that Catalan adolescents present with BD [29,30] and that relational (family and friends) and environmental (schools and social media [SM]) factors have a strong influence on the distorted perception of their own body [31]. These findings justify, firstly, the need to understand the factors that facilitate or hinder the development of a positive body image (PBI) and, secondly, to design and implement community-based interventions that promote greater SE and a PBI among Catalan adolescents.

However, most studies on SE and BI have focused on individual and group interventions, many of which address pathological situations, but lack perspectives on community-based interventions aimed at tackling these issues [32].

Based on the aforementioned, this qualitative study aims to explore and compare the views of adolescents, parents, and teachers regarding activities and possible intervention strategies to improve SE and PBI in schooled adolescents. Specifically, the research question guiding this study is: How do adolescents, parents, and teachers perceive interventions aimed at improving self-esteem and promoting a positive body image in school-aged adolescents? It is hoped that this knowledge will underpin the development of community-based intervention programmes designed to improve SE and PBI among Catalan adolescents, with effective goals [33].

## 2. Methods

### 2.1 Design

This study is framed within the constructivist inquiry paradigm with a theoretical-methodological approach that is also constructivist [34]. A generic qualitative design was employed, which is defined as one that is not guided by an explicit or established set of philosophical assumptions, such as grounded theory, phenomenology or ethnography [35]. The study was conducted and reported following the Equator Guidelines for Reporting Research using the COREQ 32-item checklist for qualitative research (See S1 File) [36].

### 2.2 Theoretical framework

The theoretical framework used in this research is based on the Health Education Model by Gómez et al. [37], a comprehensive eight-stage framework designed to enhance the effectiveness of health education interventions. The stages are: (1) situation analysis, (2) identification of health needs and problems (3), priority setting (4), goal and objective formulation (5), determination of activities and resources (6), implementation (7), evaluation, and (8) follow-up. Each stage is integral to ensuring that educational strategies are not only tailored to the specific needs of the target population but also continuously refined based on evaluation and feedback.

This study delved into stage 3, priority setting. Thus, adolescents (as the target population) and teachers and parents (as key stakeholders) helped identify the determinants and factors involved in SE and PBI. This information was structured into perceived barriers and facilitators, recognising its importance in complementing the assumptions of researchers and health professionals. Alongside the proposed intervention activities and strategies, useful foundations were provided to continue with the remaining stages for the planning of the final intervention programme.

### 2.3 Recruitment and participants

The recruitment period lasted from 11-01-2021 to 18-06-2021. Purposive sampling [38] was used to recruit the study sample comprising adolescents, teachers, and parents of adolescents from a rural school in Lleida, Catalonia, Spain.

The inclusion criteria for participants were as follows for students, teachers, and parents, respectively: 1) students in CSE and considered by their peers as someone with high SE, with good BI, and who is self-assured; 2) teachers at the school and considered by the students as references; and 3) parents of students chosen by their peers as references. Subjects who did not provide consent were excluded from the study.

The school nurse assisted with participant recruitment. To recruit adolescents, a questionnaire (ad hoc) was administered with the students asking them to write the names of three classmates that they thought had high SE, was self-assured, and had good BI, along with another questionnaire (ad hoc) asking them to write down the names of the three most important Compulsory Secondary Education (CSE) teachers from their school year; both were anonymous questionnaires. A total of 324 adolescents from all four school years responded to the questionnaires, and a definitive list was drawn up. The adolescents who received the most votes from each year group were asked to participate, as well as their parents, and the teachers who received the highest ratings.

The school nurse invited the candidates to participate in the study via telephone, and a face-to-face appointment was arranged to inform them of the aims of the study through a participant information sheet and to obtain their written informed consent (IC).

A total of nine adolescent students, seven parents of the students, and eight CSE teachers who were references for the students participated in this study (Table 1). None of the candidates refused to participate in the study.

## 2.4 Data collection

Data were collected by MPP, GTN, and EAB between July 2021 and March 2022 through semi-structured interviews. A generic interview script (for adolescents, parents and teachers) was designed based on existing literature to collect information about possible intervention strategies to improve SE and BI among adolescents (See S2 File).

Semi-structured interviews with the adolescents were conducted through video calls via Microsoft Teams, due to COVID-19 social distancing regulations at the time, and face-to-face interviews were conducted with teachers and parents at the school centre. Open-ended questions were posed, which encouraged the participants to provide additional information when responding, as relevant. Interviews with adolescents were video-recorded, and interviews with parents and teachers were audio-recorded. All interviews were recorded with the IC of the participants; the average interview duration was 30–45 minutes. The interview was finished when data saturation was reached. Notes were also taken during and after the interviews, which helped with understanding and interpreting the discourses during the analysis. Finally, all the interview data were anonymised, with an alphanumeric code assigned to each interviewee, and transcribed textually for analysis.

## 2.5 Data analysis

Thematic analysis of the data was conducted by GTN, EAB and EGM. The process commenced at the end of the first interview and involved a reflexive approach [39,40]. The transcripts were reviewed and imported into Atlas.ti 9. Inductive and deductive coding were performed to identify the meaning units of the text.

To ensure inter-coder reliability, all two researchers (GTN and EAB) and one external (EGM) independently coded the interview transcripts. The initial codes were compared, and discrepancies were discussed and resolved through consensus during team meetings. This process ensured consistency in coding and refined the coding framework. After the initial agreement, the refined framework was applied to the remaining transcripts.

**Table 1. Participants' demographics (n = 22).**

| Personal interviews | Participant ID code | Sex | Academic course[a] |
|---|---|---|---|
| **Adolescents** | A1 | Female | 1st year |
| | A2 | Female | 2nd year |
| | A3 | Male | 3rd year |
| | A4 | Female | 4th year |
| | A5 | Female | 2nd year |
| | A6 | Female | 4th year |
| | A7 | Male | 4th year |
| | A8 | Female | 4th year |
| | A9 | Female | 4th year |
| | **ID code** | **Sex** | **Age** |
| **Parents** | P1 | Female | 50 |
| | P2 | Female | 47 |
| | P3 | Female | 47 |
| | P4 | Female | 52 |
| | P5 | Male | 51 |
| | P6 | Female | 56 |
| | P7 | Female | 44 |
| **Secondary school teachers** | T1 | Female | 57 |
| | T2 | Male | 43 |
| | T3 | Male | 42 |
| | T4 | Female | 43 |
| | T5 | Female | 45 |
| | T6 | Male | 55 |
| | T7 | Female | 52 |

[a]Compulsory secondary education in Spain and Catalonia it consists of four academic years, normally between the ages 12 and 16.

All the interview transcripts were thoroughly read, and each sentence or paragraph with the same meaning was assigned a code. The codes were then grouped into subthemes and main themes (Fig 1). For instance, the following steps illustrate the coding process:

- Raw data (transcription): 'Sometimes, students just need a space to chat openly, without any judgment, and spill the beans about what's going on. They just want to feel heard' (T2)

- Code: 'Adolescents need a safe space to talk and be heard'

- Subtheme: 'Lack of family communication'

- Theme: 'Perceived barriers'

This example demonstrates how meaning units were identified, coded, and subsequently grouped into broader subthemes and themes.

The final process, including all coding and the grouping of codes and themes, was also shared among the three participants in the analysis. Discrepancies were reviewed and resolved through consensus during team meetings.

The reflexive approach enabled the research team to critically engage with the data, remaining attuned to their own biases and interpretations, ensuring that the themes accurately represented participants' perspectives.

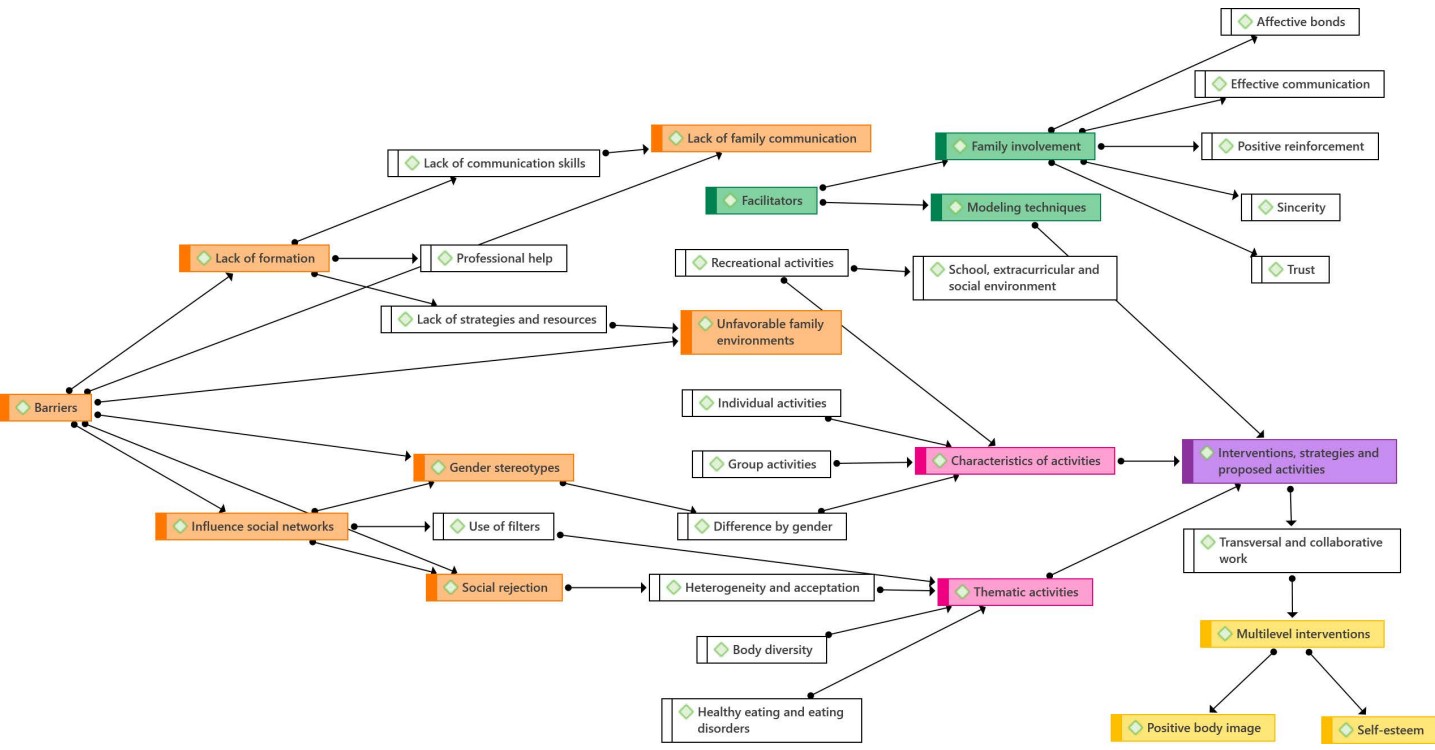

**Fig. 1. Code map.**

## 2.6 Ethical considerations

All the participants were informed that participation was voluntary and that no financial rewards would be given, based on which they all provided written IC. Information regarding the project was provided, and participants were given the opportunity to express their concerns or ask any questions pertaining to the study. Data confidentiality was assured according to Regulation (EU) 2016/679 of the European Parliament and Council of 27 April 2016. This study followed the principles of the Declaration of Helsinki and Good Clinical Practice Guidelines. This study was approved by the Clinical Research Ethics Committee of IDIAPJGol (20/154-P).

## 2.7 Rigour

Trustworthiness of the data was ensured based on the criteria of credibility, dependability, conformability, and transferability [41]. A transparent description of the steps taken since the beginning of the project was recorded. The results were based on data triangulation, as part of which individuals with different participant profiles helped generate, analyse, and interpret the data. An external coder, who independently coded and classified the material for peer review, guaranteed the rigor and reliability of the content during the thematic analysis process [42].

## 3. Results

Three main themes and 11 subthemes were identified. The most representative adolescents, teachers and parents' quotes are provided throughout this section in italics.

See Table 2 for the main themes, subthemes, and most representative quotes for quick reference.

**Table 2. Themes, subthemes and most representative quotes.**

| Themes | Subthemes | Quotes |
|---|---|---|
| Perceived barriers | Lack of family communication | 'You have to talk more with your parents. It depends on how, there are times when parents don't give confidence (...) they are too strict and the children close up (...) You have to know how to give them confidence: I am your father, I am your mother... you have to have respect, but I want to have some confidence with you, I want to know how you feel, I want to know what you think, what has happened to you today, who you like....' A8 |
| | Unfavourable family environment | 'I find that when it comes to people with low self-esteem, with a somewhat negative perception of themselves, this can often be attributed to a disorganised and inconsistent family environment' T6 |
| | Lack of training | 'All of us parents try to give as much as we can, the best advice we can. Now, maybe you don't know enough or don't have enough experience to explain things they need to know' P5 |
| | SM influence | 'On Instagram, everything looks perfect (...) people's stories, their photos, you can say look at this one, and someone who feels bad about themselves sees it and thinks they want to be like that. This affects us more and more, what people will say and their opinions' A2 |
| | Influence of gender stereotypes | 'Some girls are having a hard time because of the comments boys are making (...) saying they're fat, that they don't have breasts, that their butt is big (…) Girls suffer more than boys, and boys can be quite cruel' T3 |
| | Social rejection | 'I believe that if the group accepts you as you are, this raises your self-esteem because you see your own defects, but you notice that others don't take them into account (...) I believe at this age, it's the main factor' T2 |
| Perceived facilitators | Family involvement | 'Firstly, there should be complicity with the family to be able to talk, to explain things, not to hide from anything, not to establish taboos, right? They should be able to come home, express themselves, and vent their emotions, and from here, well, there should be good support' T3<br>'I think it has to be in both places. I also don't think that teachers can wash their hands of all this and say, "No, you do this at home." I believe that at school, you can work on many things, and that's good, because the parents are not there' P2 |
| | Modelling as a strategy | 'Explain to others how you developed such high self-esteem and how you accept yourself, and then look for defects in yourself so that you can explain to others how you are aware of these defects and yet maintain your high self-esteem. That they have references that teach them that the image is not the primary thing in this life (...) People who have suffered, people who have gone through a process of self-knowledge and have felt, in quotes, mistreated by society for not being equal, the stereotypes that society imposes.(...) if they can see real examples, they will understand it much more than when conveyed just in words' T3 |
| Activities and intervention strategies | Characteristics of activities | 'In a group, they wouldn't open up as much, and they wouldn't be as honest. If you go individually, I suppose you would help them more. I guess you're afraid of expressing yourself in front of everyone´ A2<br>'I think that primary school is a good time because that's when they are perhaps more receptive (...) is where it has a greater impact, and in secondary school, you have to keep working, but in a more subtle and gradual way´ T5<br>'Local associations, those places they go to because they want to (…) and in sports as well, and in extracurricular activities (…) If they need collaboration from the city council, it's provided, but organizing things for them, I think, would be a mistake (…) They have to organize it themselves. It has to be their initiative, I think´ P2 |
| | Proposed themes | 'You really have to accept yourself as you are, because if not, you'll suffer your whole life saying 'I want to be like this,' and in the end, you have to end up accepting yourself as you are (...) Don't compare yourself to other people, if you can change something, then go ahead and try to feel good about yourself, because self-esteem and self-love are very important´ A6 |
| | Types of activities | 'Surely there are workshops that can be fun and at the same time make you reflect. There are probably many specialists who know about these things and can conducting a workshop, for example, at school´ A8<br>'For me, it's a subject (...) about coexistence, values... everything, at school´ P7<br>'From all subjects, it can be conveyed. Sometimes it feels like it's only the tutor who has to do it or the teacher of culture and ethical values or religion who has to do it. But it's not true, in the end, we all are providing a model of image´ T2 |

## 3.1 Theme 1: Perceived barriers

### 3.1.1 Lack of family communication.
Teachers acknowledged that adolescents need environments where they can talk and feel heard, although the adolescents confessed that physical appearance was not important in the family environment and that they did not discuss this topic with their parents. Some parents believed that talking too much to their children about BI and associated problems could be counterproductive. Additionally, both parents and teachers perceived that adolescents did not listen to them, with their valuing their friends' opinions more.

> *'Sometimes, students just need a space to chat openly, without any judgment, and spill the beans about what's going on. They just want to feel heard' T2*

> 'She thinks that if we tell her "you're thin," we're telling her because I'm her mother and we want her to eat (...) In adolescence it's more important what friends say or what someone who doesn't know them at all says than their parents' P7

The adolescents confirmed that they sought refuge in their friends due to a feeling of lack of communication and trust in the family environment, expressing that they would trust their parents more if they improved their communication skills.

> 'You have to talk more with your parents. It depends on how, there are times when parents don't give confidence (...) they are too strict and the children close up (...) You have to know how to give them confidence: I am your father, I am your mother... you have to have respect, but I want to have some confidence with you, I want to know how you feel, I want to know what you think, what has happened to you today, who you like....' A8

**3.1.2 Unfavourable family environment.** Teachers associated students with low SE or BD with unfavourable family environments. According to the teachers, parents with insecurities, fears, hypercritical attitudes, or high demands instilled insecurity, fear, and low SE in their children. They also reported examples of adolescents receiving negative comments from their parents regarding their BI, and encouraging unhealthy exercise and eating habits.

> 'I find that when it comes to people with low self-esteem, with a somewhat negative perception of themselves, this can often be attributed to a disorganised and inconsistent family environment' T6

> 'If you have a father or a mother who is very insecure or afraid of everything, transmits this fear and insecurity. The child tends to be the same, not always, but in the majority of cases (...) Sometimes they tell me... my dad or my mom tells me I'm fat, that I shouldn't eat' T7

**3.1.3 Lack of training.** Teachers acknowledged difficulties in managing complex situations in the classroom, leading them to request training to learn about strategies, management resources, and activities to promote healthy SE and BI. Additionally, they believed that the initiatives carried out in the classroom (See S3 File) should be reinforced at home, deeming it crucial to train parents to achieve collaborative work. Parents confirmed the need for training.

> 'Parent education plays a significant role. As a teacher, what I've found is that parents are acting as pedagogues with their children to a lesser extent, that is, they subsist, they have a lot of doubts, and don't know how to handle it (...) because we encounter many situations where we don't know how to react, situations which are complex (...) If there is good training, and this is lacking, I think the rest is also easier to manage' T2

> 'All of us parents try to give as much as we can, the best advice we can. Now, maybe you don't know enough or don't have enough experience to explain things they need to know' P5

Given the lack of training, both teachers and parents believed that it was most appropriate to seek help from educational psychologists, psychologists from the educational institution, and primary care professionals. For parents, consulting a doctor represented a quick solution to the problem.

> 'If I can't help her anymore, I would take her to a doctor quickly because the solution is a doctor and fixed quickly´ P3

**3.1.4  SM influence.**  Teachers perceived that adolescents are dissatisfied with themselves because of the physical and psychological changes they experience. According to the teachers, society promotes beauty standards through SM, which is saturated with inappropriate values and messages. Additionally, they believed that adolescents were unaware of the influence this has on their physical and mental well-being.

> *'Always this background of social media, music... all of this that affects adolescents (...) negatively affects them a lot, and they don't realise it' T3*

> *'If I have a YouTuber as a role model, and a YouTuber can make a living, a very good one, by making four videos, why should I make an effort?' T2*

In contrast, adolescents were aware that SM led them to compare themselves with others based on certain beauty standards, thereby increasing their BI concerns and BD. They believed that it was challenging to prevent the risks of SM, although parents insisted on raising awareness among adolescents about how filters are used in photography to 'enhance' the body representations seen online.

> *'On Instagram, everything looks perfect (...) people's stories, their photos, you can say look at this one, and someone who feels bad about themselves sees it and thinks they want to be like that. This affects us more and more, what people will say and their opinions' A2*

> *'Explain clearly what filters are and conduct many talks about internet-related topics, explaining the harm that a word can do to a person, which can destroy their life' P7*

**3.1.5  Influence of gender stereotypes.**  Adolescents and teachers believed that boys possessed better BI and greater self-confidence and placed less importance on physical appearance. However, girls believed that physical appearance was also a concern for boys, even if not explicitly expressed. Teachers acknowledged an increasing number of cases where boys felt uncomfortable about their physical appearance, which negatively impacted their involvement in collective activities.

> *'Girls, more than boys, tend to obsess more about their appearance and chase an ideal (...) Boys might not talk about it, but they feel it inside too' A4*

> *'Sometimes it seemed like this was more of a girls' issue, but now we find it more in boys as well' T2*

> *'Some boys, if they are chubby or have a belly, they are also ashamed of engaging in certain activities in public in front of others (...) They wear looser clothes, such as sweatshirts, even when it's hot, or they don't even want to take off their coats in the winter' T7*

Some teachers contended that girls were more affected by derogatory comments regarding their physical appearance. Girls deemed it necessary to engage in activities that promoted increased participation and awareness among boys. Furthermore, they proposed implementing separate gender-specific activities and a collective discussion to juxtapose viewpoints.

> *'Some girls are having a hard time because of the comments boys are making (...) saying they're fat, that they don't have breasts, that their butt is big (…) Girls suffer more than boys, and boys can be quite cruel' T3*

*'If we conduct a workshop sometimes only for girls and only children? (...) Sometimes divided, we see the female perspective and the male perspective, and then we come together, and everyone learns more'* A8

**3.1.6 Social rejection.** According to teachers, group acceptance would contribute to high SE. The majority of adolescents dressed similarly to feel part of the group, as they believed that both clothing style and personality traits influenced their social acceptance or rejection. Additionally, there was evidence of a minority of adolescents who did not identify with the group and tended to isolate themselves. Teachers hope that adolescents learn to respect and accept differences. To achieve this, the heterogeneity among students should be discussed across various subjects, involving the entire teaching staff.

*'I believe that if the group accepts you as you are, this raises your self-esteem because you see your own defects, but you notice that others don't take them into account (...) I believe at this age, it's the main factor'* T2

*'I want him to learn to accept and value those who are not like him. That's my goal'* T7

## 3.2 Theme 2: Perceived facilitators

**3.2.1 Family involvement.** Parents and teachers acknowledged that the family plays a crucial role in the development of SE. For parents, having high SE and a healthy BI is important for life, and they believe that these aspects should be addressed within the school environment. Teachers believe that adolescents' social environment influences their SE and BI, both relying on the values instilled by the family since childhood. They believed that parents should encourage effective communication with positive reinforcement, trust, and sincerity.

*'The family, in principle, is the first place. A solid family environment should be a certain degree of solidity and coherence with the parents or those who take charge. A certain coherence is also necessary in their attitudes, which can transmit values'* T6

*'I think it has to be in both places. I also don't think that teachers can wash their hands of all this and say, "No, you do this at home." I believe that at school, you can work on many things, and that's good, because the parents are not there'* P2

*'Firstly, there should be complicity with the family to be able to talk, to explain things, not to hide from anything, not to establish taboos, right? They should be able to come home, express themselves, and vent their emotions, and from here, well, there should be good support'* T3

Parents confirmed and emphasised the importance of emotional bonds within the family and behaviours that reinforce positive attitudes towards adolescents, although they believed it was necessary to establish clear behavioural guidelines to instil a sense of security.

*'I really like to ask and ask. I know that sometimes they won't tell me everything, but a mother with a child or a father has to have a lot of trust with their children'* P3

*'It is to provide positive reinforcement and make them see what they are worth, even if at school they are not made to see it or they do not fully see it, encouraging them to look for things they can excel in'* P2

*'I have always given them very clear guidelines on what to do in many situations (...) I believe all of this contributes to self-esteem, making them feel secure'* P4

**3.2.2 Modelling as a strategy.** Parents and adolescents believed that gaining closer insights into the experiences of other teenagers or individuals with low SE or BD could assist them in reflecting on and reconsidering their own negative thoughts.

> *'How do you achieve this self-esteem? What has happened to you to have this self-esteem? Because if I have self-esteem now, it does not mean that I had it five years ago, or that I have not been bullied or that I've always been the centre of attention. Each person has a way of behaving, and this behaviour depends on what has happened to them'* A8

> *'I would conduct talks with adolescents who have self-esteem, who have gone through tough times, who have been hospitalised due to self-esteem issues... there are conditions like depression and other illnesses because of this. Real-life stories, people who can come and speak'* P7

For teachers, exposing adolescents to real-life cases could bring them closer to the sociocultural reality, enhancing their understanding of these issues.

> *'Explain to others how you developed such high self-esteem and how you accept yourself, and then look for defects in yourself so that you can explain to others how you are aware of these defects and yet maintain your high self-esteem. That they have references that teach them that the image is not the primary thing in this life (...) People who have suffered, people who have gone through a process of self-knowledge and have felt, in quotes, mistreated by society for not being equal, the stereotypes that society imposes.(...) if they can see real examples, they will understand it much more than when conveyed just in words'* T3

## 3.3 Theme 3: Activities and intervention strategies

**3.3.1 Characteristics of activities.** Adolescents and teachers believed that individual activities could be more beneficial than group activities for improving SE, especially for adolescents concerned about their BI or those facing social rejection.

> *'Depending on the activity, but group activities often make people who have some issues feel terrible (...) group activities, not at all´* T6

> *'In a group, they wouldn't open up as much, and they wouldn't be as honest. If you go individually, I suppose you would help them more. I guess you're afraid of expressing yourself in front of everyone´* A2

According to teachers and parents, the primary education stage (6 – 12 years of age) is the most suitable period for conducting activities, as they believe that SE is established during early adolescence.

> *'Self-esteem is not at 16. Self-esteem is shaped at 9 (...) If you do it outside the centre, they won't come´* P7

> *'I think that primary school is a good time because that's when they are perhaps more receptive (...) is where it has a greater impact, and in secondary school, you have to keep working, but in a more subtle and gradual way´* T5

Additionally, they considered the educational institution to be the best place for these activities, given the significant amount of time spent there. These explanations were supported by

the majority of adolescents, although some expressed the opinion that discussions within the educational setting might be ineffective, suggesting alternative environments, including SM.

*'Sometimes what is said at school is like wow! It has been said at school! It's something important, you know? When you're young, the information really gets to you, and school is like going to university when you are 6 or 8 or 12 years old´* A8

*'In high school no one would pay much attention. They give talks and people end up doing what they want, they don't listen (...) I believe it's better through social media´* A1

*'Doing some kind of talk for those who are more interested, but not at the institute!´* A7

Some parents supported the use of social, sports, and extracurricular settings. Parents believed that successful activities should be proposed by the adolescents themselves, so they should be provided with the necessary resources and social support to carry them out. They also believed that including family members from various generations in leisure and recreational activities could be enriching and bring diverse perspectives.

*'What I would do is promote cultural entities and associations like the scouts, where everyone can participate (...) where everyone works equally, and I don't know... as I would put it, socialize´* P5

*'Local associations, those places they go to because they want to (…) and in sports as well, and in extracurricular activities (…) If they need collaboration from the city council, it's provided, but organizing things for them, I think, would be a mistake (…) They have to organize it themselves. It has to be their initiative, I think´* P2

*'I believe that activities should always involve more than one generation, but at the same time, they should be creative and fun (…) You enrich yourself when you have other perspectives´* P1

**3.3.2  Proposed themes.**  The majority of adolescents believed that future activities aimed at improving SE and BI should focus on normalising the understanding that there are all kinds of bodies, all of which are equally valid and can be healthy, without stigmatising or discriminating against individuals based on their appearance. They considered that addressing the effects of SM from early adolescence could prevent issues associated with BD and requested information on healthy eating and ED to increase social awareness. The adolescents deemed accepting oneself and self-love crucial for liberation from suffering.

*'There's always talk about models being tall, having a 90-60-90 figure, and they have to be perfect (…) People give a very wrong idea about this, and on social media, you always see a certain type of person: the hot girl with a very big butt, and the hot guy is the one who is muscular and does abs (...) When you're young, especially if you don't talk about this with anyone and no one can help you, it's normal that it affects you quite a bit´* A8

*'You really have to accept yourself as you are, because if not, you'll suffer your whole life saying 'I want to be like this,' and in the end, you have to end up accepting yourself as you are (...) Don't compare yourself to other people, if you can change something, then go ahead and try to feel good about yourself, because self-esteem and self-love are very important´* A6

*'Most of us who end up with eating disorders come from people's comments (…) People give their opinions, and in reality, they don't know anything and don't realize how a comment can affect another person (...) The whole concept of miracle diets, all of this needs*

*to be explained. Because I've also been a person who didn't have the concepts of healthy eating´ A5*

### 3.3.3 Types of activities.

Adolescents admitted to receiving informative talks in the educational environment on gender perspectives, gender-based violence, sexually transmitted diseases, and substance dependencies. However, they called for information on how to improve SE and activities with continuity over time. Therefore, they proposed workshops in educational institutions with recreational dynamics to work on SE and promote reflection.

*'Usually, they come to give us many talks about sexual diseases or drugs, but never about self-esteem´ A4*

*'Surely there are workshops that can be fun and at the same time make you reflect. There are probably many specialists who know about these things and can conducting a workshop, for example, at school´ A8*

However, parents and teachers considered it more appropriate for adolescents to have a subject on values and emotional education to encourage reflection and enhance social and emotional skills. Some teachers believed that these topics should not be addressed in a single subject but rather integrated throughout the school curriculum.

*'Emotional education is about working on the emotions of the students, firstly identifying which emotions exist, and secondly, figuring out how these emotions can be managed´ T2*

*'For me, it's a subject (...) about coexistence, values... everything, at school´ P7*

*'From all subjects, it can be conveyed. Sometimes it feels like it's only the tutor who has to do it or the teacher of culture and ethical values or religion who has to do it. But it's not true, in the end, we all are providing a model of image´ T2*

## 4. Discussion

This qualitative study has revealed numerous barriers to the effective implementation of interventions aimed at improving SE and PBI among adolescents. Among the highlighted barriers were those related to an unfavourable and uncommunicative family environment, which was associated with students experiencing BD. On the one hand, adolescents expressed a desire for more communication with their parents, while parents and teachers perceived that adolescents were not listening to them and valued their friends' opinions more. This is consistent with the literature, which highlights that adolescents during this stage of life tend to gravitate towards their peers and distance themselves from adults, prioritising the influence of their friends over that of authority figures [32]. Quality family relationships are crucial during adolescence, as several studies suggesting that BD among adolescents is influenced by their relationship with their parents [43]. Adolescents who experience positive family relationships feel more secure and exhibit less BD [44]. This renders them less susceptible to sociocultural influences, making it less likely for them to internalise the beauty ideals propagated by SM and feel the need to conform to these ideals for societal acceptance [45].

Another barrier identified was the use of SM, leading adolescents, regardless of their gender, to compare themselves with others based on beauty standards and increasing their BI concerns and BD, consistent with previous studies [46,47]. Individual factors such as the tendency for social comparison [48,49] and media literacy level [49,50] may moderate this effect. While BD was evident across genders, generally, boys had better BI, and girls suffered

more derogatory comments about their physical appearance, negatively influencing them and affecting their participation in collective activities. These differences could be explained by the gender-based vulnerability to cultural influences and the current beauty ideals promoted by SM, which value thinness in women and a strong and muscular body in men [51,52]. Furthermore, exposure to teasing, defined as the experience of being the target of provocation or a form of harassment for individuals with physical or behavioural characteristics distinguishing them from the majority, often with a negative connotation [53], could explain poorer BI among girls. Teasing has been strongly associated with BD [53] and inappropriate weight loss behaviours such as purging and restrictive diets [54,55]. Health behaviours have been linked to body appreciation and positive body image and are closely connected to sociocultural aspects [32].

Regarding barriers related to teacher and parent education, this study reveals the need to promote the acquisition of skills and competencies to enhance higher SE and PBI among adolescents. Modelling was suggested as a facilitating technique, as through observing the actions of others, observers can acquire cognitive skills and new behaviour patterns [56]. This technique generates emotional activation that could help adolescents reassess their negative thoughts, increase their self-efficacy, and moderate the effects of social comparison [56]. Therefore, it could be incorporated as a social learning technique in group activities aimed at adolescents in the various proposed settings (school, extracurricular, and social) for future interventions.

The involvement and engagement of the family in activities emerged in this study as a relevant facilitator, as the behaviours and interactions of adults become models for adolescents' own behaviour [57]. From early life, the family constitutes the basic reference nucleus, bringing together all reference models [57]. Later, the educational institution provides new models, with the teacher being the fundamental model through which adolescents visualise different and even contradictory behaviours than those of the family. Accordingly, this study reveals the need to include collaborative family participation in school activities. However, another significant source of models corresponds to mass media, comprising significant figures for adolescents; the likelihood of their behaviour being imitated by adolescents is high [57]. In this study, when the figure acting as a model presents with low SE, BD, or what could be risky behaviours for adolescents, individual interventions have been proposed. However, parents and teachers considered it more appropriate to seek professional help from the educational or primary care centre, as they have established diagnostic and treatment care pathways.

Another important finding was regarding the proposed interventions to improve SE and BI. The importance of activities in schools from early childhood and with active family participation, through individual and collective recreational activities of interest to adolescents, was emphasised. These interventions should ideally focus on respecting and accepting different body shapes, while considering gender differences. All of these aspects could be addressed through modelling or other techniques (cognitive restructuring, relaxation training, and interpersonal problem-solving) to improve social skills and strengthen behaviours for social interaction [58]. Moreover, peer support has been identified as a key element in interventions aimed at increasing body appreciation and self-esteem among adolescent girls [32]. Our results suggest the incorporation of a specific subject about emotional intelligence into the educational curriculum. However, it seems that the best option would be to implement an integrative learning project that includes all subjects and the respective teachers. This would allow us to intervene effectively, supporting adolescents as they develop their values and behaviour patterns through daily actions; improving the way they relate to others; and helping them develop conflict resolution strategies, the ability to identify and express emotions, and empathic behaviours—all of which are aspects crucial for their development [59].

This study point to the importance of developing and implementing interventions to improve SE and BI among adolescents using a structured theoretical framework, such as the Socio-Ecological Model (SEM). The SEM recognises that an individual's behaviour is shaped through complex interactions and relationships of influence based on interpersonal, organisational, community, and public policy factors [60]. Under this premise and considering that this findings highlights the importance of incorporating specific aspects into activities and intervention strategies that consider sociocultural and gender differences, both in educational settings and other environments of adolescent socialisation, there is a need to implement multi-level interventions (MLIs). For example, school policies could integrate body image and self-esteem awareness into the curriculum, addressing these topics across different subjects. Additionally, some of these students could lead activities or workshops aimed at their classmates and peers from other year groups to address questions and encourage discussions on these issues. Public health initiatives could involve community programmes that include both adolescents and adults, challenging harmful beauty standards, promoting body diversity, and encouraging healthier perceptions of body image, among other topics. By incorporating these strategies, MLIs would offer a holistic approach that engages the entire population, not just adolescents, but also families, educational institutions, and broader local community settings [61,62].

Future studies should use the facilitators identified in this study to design interventions aiming to increase adolescents' SE and BI while promoting good physical and mental health during their transition to adulthood. These interventions could be embedded in both school systems and public health campaigns, fostering a supportive environment for adolescents' mental well-being.

## 4.1 Strength and limitations

One of the strengths of this study was the sample with different participant profiles, ensuring maximum variation in conceptual meanings. Additionally, theoretical sampling was used, which may allow for greater transferability and applicability to other contexts, providing external validity [63].

The main limitations of this study are as follows. First, the interview schedules were interrupted due to the COVID-19 pandemic, which complicated the data collection and analysis processes. Secondly, the use of purposive sampling combined with peer nomination may have introduced potential biases, such as the influence of group dynamics or social relationships. Efforts were made to mitigate these biases by clearly explaining the nomination process to participants. Thirdly, there was a disproportionate sex representation among the adolescent participants, with only two males of the nine adolescents interviewed. Since the recruitment of adolescents was based on their references within the group, the overrepresentation of girls could reflect a social reality where women more clearly experience and express the societal standards related to body image and self-esteem, while men tend to go unnoticed. Although the results indicate adolescents recognize different perceptions between boys and girls, this imbalance in the sample should be taken into account when interpreting the findings. Finally, this study focused on a school in a rural area of Catalonia, which limits the generalisability of the findings; therefore, the results must be interpreted within this context. Socio-cultural and economic factors in rural settings can differ from those in urban areas, affecting how adolescents perceive, and experience issues related to self-esteem and body image. Future research could explore similar interventions in urban schools, where greater socio-economic diversity and a more heterogeneous population may present distinct challenges and opportunities. Comparing findings from both rural and urban settings would provide a deeper understanding of how these contextual factors influence adolescents' well-being and help develop tailored intervention strategies adapted to the specific environment.

## 5. Conclusions

This study highlights various barriers to implementing effective interventions for improving SE and BI among adolescents. Based on the findings, we propose strategies that involve the educational community, parents, families, and society as a whole. Future intervention measures should focus on reinforcing identified facilitators with modelling techniques, for example, and addressing the barriers, emphasizing the need for multi-level policies and public actions at local and governmental levels. We urge health policy planners and managers to promoting inclusive educational practices, supporting parental involvement, and fostering community-driven initiatives to enhance adolescent well-being. By addressing these key areas, we can more effectively tackle the challenges surrounding SE and BI among adolescents.

## Supporting information

**S1 File. Checklist COREQ.**
(DOCX)

**S2 File. Guide interview.**
(DOCX)

**S3 File. Statements by teachers about examples of classroom activities.**
(DOCX)

## Acknowledgments

We would like to thank Laura Marimon Gabernet and Marta Tolosa Fortuny of the Primary health care in Lleida (Catalonia, Spain) for their collaboration.

## Author contributions

**Conceptualization:** Eva Artigues-Barberà, Mercè Pollina-Pocallet.

**Data curation:** Eva Artigues-Barberà, Glòria Tort-Nasarre, Mercè Pollina-Pocallet, Yvonne Ferrer Suquet.

**Formal analysis:** Eva Artigues-Barberà, Glòria Tort-Nasarre, Ester García-Martínez.

**Funding acquisition:** Eva Artigues-Barberà, Mercè Pollina-Pocallet.

**Investigation:** Eva Artigues-Barberà, Glòria Tort-Nasarre, Mercè Pollina-Pocallet.

**Methodology:** Eva Artigues-Barberà, Glòria Tort-Nasarre, Mercè Pollina-Pocallet.

**Validation:** Ester García-Martínez.

**Visualization:** Ester García-Martínez.

**Writing – original draft:** Eva Artigues-Barberà, Glòria Tort-Nasarre, Mercè Pollina-Pocallet, Ester García-Martínez.

**Writing – review & editing:** Eva Artigues-Barberà, Glòria Tort-Nasarre, Mercè Pollina-Pocallet, Yvonne Ferrer Suquet, Aleix Ayats Pallés, Olivía Guasch Niubó, Marta Ortega-Bravo, Ester García-Martínez.

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
