## [Decision Letter · Decision Letter 0]

29 Dec 2024

PONE-D-24-42336Key factors in supporting adolescents to achieve high self-esteem and a positive body image: A qualitative community-based studyPLOS ONE

Dear Dr. García Martínez,

Thank you for submitting your manuscript to PLOS ONE. After careful consideration, we feel that it has merit but does not fully meet PLOS ONE’s publication criteria as it currently stands. Therefore, we invite you to submit a revised version of the manuscript that addresses the points raised during the review process.

We look forward to receiving your revised manuscript.

Kind regards,

Mukhtiar Baig, Ph.D.

Academic Editor

PLOS ONE

Journal requirements: When submitting your revision, we need you to address these additional requirements. 1. Please ensure that your manuscript meets PLOS ONE's style requirements, including those for file naming. The PLOS ONE style templates can be found at https://journals.plos.org/plosone/s/file?id=wjVg/PLOSOne_formatting_sample_main_body.pdf and https://journals.plos.org/plosone/s/file?id=ba62/PLOSOne_formatting_sample_title_authors_affiliations.pdf. 2. Thank you for stating the following financial disclosure:  [PICARD award of Provincial Council of Lleida (code 4R23/360).].  Please state what role the funders took in the study.  If the funders had no role, please state: ""The funders had no role in study design, data collection and analysis, decision to publish, or preparation of the manuscript."" If this statement is not correct you must amend it as needed. Please include this amended Role of Funder statement in your cover letter; we will change the online submission form on your behalf. 3. We note that you have indicated that there are restrictions to data sharing for this study. For studies involving human research participant data or other sensitive data, we encourage authors to share de-identified or anonymized data. However, when data cannot be publicly shared for ethical reasons, we allow authors to make their data sets available upon request. For information on unacceptable data access restrictions, please see http://journals.plos.org/plosone/s/data-availability#loc-unacceptable-data-access-restrictions.  Before we proceed with your manuscript, please address the following prompts: a) If there are ethical or legal restrictions on sharing a de-identified data set, please explain them in detail (e.g., data contain potentially identifying or sensitive patient information, data are owned by a third-party organization, etc.) and who has imposed them (e.g., a Research Ethics Committee or Institutional Review Board, etc.). Please also provide contact information for a data access committee, ethics committee, or other institutional body to which data requests may be sent. b) If there are no restrictions, please upload the minimal anonymized data set necessary to replicate your study findings to a stable, public repository and provide us with the relevant URLs, DOIs, or accession numbers. Please see http://www.bmj.com/content/340/bmj.c181.long for guidelines on how to de-identify and prepare clinical data for publication. For a list of recommended repositories, please see https://journals.plos.org/plosone/s/recommended-repositories. You also have the option of uploading the data as Supporting Information files, but we would recommend depositing data directly to a data repository if possible. Please update your Data Availability statement in the submission form accordingly.

Reviewers' comments:

Reviewer's Responses to Questions

**Comments to the Author**

1. Is the manuscript technically sound, and do the data support the conclusions?

Reviewer #1: Yes

Reviewer #2: Partly

Reviewer #3: Yes

2. Has the statistical analysis been performed appropriately and rigorously?

Reviewer #1: Yes

Reviewer #2: N/A

Reviewer #3: Yes

3. Have the authors made all data underlying the findings in their manuscript fully available?

Reviewer #1: Yes

Reviewer #2: Yes

Reviewer #3: Yes

4. Is the manuscript presented in an intelligible fashion and written in standard English?

Reviewer #1: No

Reviewer #2: Yes

Reviewer #3: Yes

5. Review Comments to the Author

Reviewer #1: Manuscript Review

Dear Editor,

Thank you for sending the manuscript titled “Strategies to Improve Self-Esteem and Body Image Among Adolescents: Barriers and Facilitators to Intervention” for review. Here are the details of each aspect of the paper and my suggestions:

1. Rigor: The manuscript presents a qualitative study that carefully examines barriers and facilitators in improving self-esteem (SE) and body image (BI) among adolescents. It appears well-structured, with a clear presentation of themes derived from participant interviews. The qualitative methodology is appropriate for exploring personal experiences and perspectives. However, the description of the data collection methods could benefit from more detail. For example, elaborating on how participants were selected and how interviews were conducted would enhance the rigor by providing clearer context for the findings.

2. Coherence: The manuscript is coherent, with logical flow and organization. The themes derived from adolescents, parents, and teachers are clearly presented, highlighting the different perspectives on SE and BI interventions. The discussion effectively ties the qualitative findings back to the existing literature, enabling readers to understand the relevance and implications of the results. Nevertheless, some sections, especially the proposals for future interventions, could be more directly linked back to the barriers identified earlier in the document.

3. Scientific Integrity: The manuscript maintains a good level of scientific integrity. Participants seem to have been treated with respect, and the concerns about ethics are briefly acknowledged, e.g., regarding informed consent. However, providing more explicit details on ethical approval processes, participant anonymity, and potential biases in the research design would contribute to the transparency of the research. It may also be helpful to reflect on potential researcher biases in data interpretation.

4. Ethical Considerations: The manuscript acknowledges the sensitive nature of discussing issues like body image and self-esteem. However, ethical considerations warrant more thorough discussion, especially with regard to adolescent participation. More information about the age range of participants, parental consent, and how sensitive topics were handled during interviews would enhance the ethical robustness of the study. A discussion of how the researchers supported participants who may have experienced distress from topic discussions could also be considered.

5. Comments and Suggestions:

• Data Collection Details: IT would be good to offer more detail about the methods used for data collection, including sampling methods, interview guide structure, and how interviews were analyzed (e.g., thematic analysis approach).

• Linking Findings to Interventions: It is imperative to strengthen the reasoning behind proposed interventions by linking them back to specific barriers more explicitly. Discuss how each suggested intervention directly addresses identified barriers.

• Diversity of the Sample: While diversity is mentioned as a strength, discussing how it may affect the transferability of findings to urban settings or other cultural contexts could improve the discussion.

• Limitations and Future Research: While limitations are acknowledged, it would be valuable to suggest specific areas for future research based on the findings and limitations identified in this study.

Conclusion: Overall, this manuscript shows promise for publication after implementing the suggestions for enhancing rigor, coherence, and scientific integrity. For submission to a peer-reviewed journal, revising with attention to ethical considerations and linking recommendations to identified barriers will strengthen the work and enhance its contribution to the field of adolescent mental health and educational interventions.

The manuscript you provided outlines a qualitative study focused on improving self-esteem (SE) and body image (BI) among adolescents, identifying barriers and proposing interventions. While it contains valuable insights, it also has several shortcomings that can be improved upon. Below are some of the key shortcomings along with suggestions for enhancement:

Shortcomings of the Manuscript

1. Lack of Clarity and Structure:

o The manuscript is densely packed with ideas without clear transitions or headings that guide the reader. For instance, the transition between quotes, themes, and discussion points is often abrupt.

o Suggestion: Organize the manuscript into well-defined sections with headings and subheadings that reflect the key themes being discussed (e.g., introduction, methodology, findings, discussion, conclusion). This would enhance readability and provide a clearer flow of ideas.

2. Insufficient Detail on Methodology:

o The methodology section lacks specificity regarding how participants were selected, the criteria for inclusion, and the data analysis approach.

o Suggestion: Expand on the methodology by providing detailed descriptions of participant recruitment, demographic information, data collection methods, and the analytical framework used. This will allow readers to better understand the study's context and rigor.

3. Inconsistent Terminology:

o Terms such as "body diversity," "body disappointment" (BD), and "self-acceptance" are used but not consistently defined or explained.

o Suggestion: Define key terms and keep their usage consistent throughout the manuscript. This will help to avoid confusion and ensure that readers understand the concepts being presented.

4. Conclusions Lack Cohesion:

o The conclusions drawn do not seem to effectively summarize the main findings of the research or provide clear implications based on the results.

o Suggestion: Revise the conclusion to more closely reflect the primary findings and their implications for practice. Consider including clear recommendations for stakeholders (e.g., educators, parents, community leaders) based on the identified barriers and facilitators.

5. Limited Discussion of Social and Cultural Context:

o While the manuscript touches on sociocultural influences, it does not delve deeply into how differing cultural backgrounds might affect adolescents' experiences of SE and BI.

o Suggestion: Incorporate a discussion on how different cultural contexts may shape adolescents' perceptions and challenges regarding body image and self-esteem. This could broaden the applicability and relevance of your findings.

6. Neglect of Adolescent's Voices:

o The adolescents' quotes are valuable but lack a more profound contextual interpretation or analysis, leaving their voices somewhat isolated.

o Suggestion: Provide thematic analysis to draw connections between the quotes and broader themes within the literature. This will help to illustrate how the viewpoints of adolescents fit within established frameworks on body image and self-esteem.

7. Call for Future Research is Underdeveloped:

o The section on future research is brief and lacks depth regarding the specific areas of study that could be beneficial.

o Suggestion: Elaborate on future research avenues and propose specific questions or hypotheses that could further investigate the topics of SE and BI among adolescents.

Overall Recommendations for Improvement

• Engage in rigorous peer review: Before publication, ensure the manuscript is reviewed by peers familiar with adolescent psychology, to identify areas needing clarification or argumentation.

• Use visuals: Consider adding tables or figures to summarize data or model the proposed interventions visually.

• Incorporate feedback mechanisms: Post-study, include ways for participants to provide feedback on the workshops or interventions. This could add further depth to the ongoing research.

• Interdisciplinary Input: Collaborate with experts in psychology, sociology, and education during revisions for a more rounded approach to the manuscript's content.

By addressing these shortcomings and following the suggestions for improvement, the manuscript can provide clearer insights, stronger arguments, and a more compelling case for interventions aimed at improving self-esteem and body image among adolescents.

Reviewer #2: For question 1- The manuscript is technically sound in its design as a qualitative study, employing appropriate methods such as semi-structured interviews and thematic analysis. However as a qualitative study, it lacks quantitative data, controls, and replication, which are critical for generalizability and for demonstrating statistical significance of the conclusions. Thus, while the qualitative insights are valuable, the manuscript does not fully meet the criteria for "rigorous experimentation" as typically defined in technical research. As for those reasons manuscript is partly technically sound.

For question 2- The manuscript is a qualitative study and does not involve statistical analysis. Instead, it relies on thematic analysis to interpret qualitative data. Statistical analysis is not applicable for this manuscript.

For question 3- The manuscript states that data are available from the corresponding author upon reasonable request. So authors did not made all data underlying finding fully available.

For question 4- The manuscript is written in clear and standard English, with no significant typographical or grammatical errors that would create issue for understanding.

Reviewer #3: The manuscript addresses a topic that is extremely relevant to the fields of health and education. Overall, it is very well designed and presented, with all the elements required by the journal. Here are my considerations and suggestions.

Introduction and goal:

The introduction is robust and presents all the concepts needed to discuss the data. The goal of the study is adequately outlined.

Method

In view of the study participants, the constructivist theoretical-methodological approach is appropriate.

It presents all the necessary elements that justify conducting the study. I would highlight the use of data triangulation as a strong point.

Results

The categories of analysis are well delineated and consistent with the study proposal.

Discussion

The discussion is well laid out, focusing initially on the barriers. It then points out important elements that should be considered in the intervention, based on the results presented.

However, my consideration lies in one very significant result. In the results, “both parents and teachers perceived that adolescents did not listen to them, with their valuing their friends’ opinions more”. This opinion is also shared by adolescents.

The literature shows that adolescents in this period of life tend to get closer to their peers and distance themselves from adults. I therefore believe that this result should be taken into account in discussions.

As for the proposed design of the intervention, the socio-ecological model proposed as the intervention's methodological strategy seems to me to be appropriate, as it allows for an interconnection between the different contexts of the adolescents' lives. I also suggest considering peer support, as pointed out earlier.

Limitation

I suggest that the authors consider the disproportionate number of boys and girls as a limitation of the study. The results themselves show that adolescents understand that there are different perceptions between the sexes.

I congratulate the authors on this very significant study for the academic and educational community.

6. PLOS authors have the option to publish the peer review history of their article (what does this mean? ). If published, this will include your full peer review and any attached files.

**Do you want your identity to be public for this peer review?** For information about this choice, including consent withdrawal, please see our Privacy Policy .

Reviewer #1: **Yes: ** Muneeb A. Faiq

Reviewer #2: **Yes: ** Md Shafayat Hossain, PhD, Postdoc (UoB, UK)

Reviewer #3: No

---

## [Author Response · Author response to Decision Letter 1]

21 Jan 2025

Dear Editor of Plos One,

I am pleased to resubmit for your consideration the manuscript PONE-D-24-42336, titled ‘Key factors in supporting adolescents to achieve high self-esteem and a positive body image: A qualitative community-based study’.

We are very grateful for the excellent suggestions and comments from the reviewers. We have carefully considered all the suggestions and comments and made revisions accordingly.

Below, we have included all the comments and suggestions from the academic editor and the three reviewers, as well as the authors' responses to each of them. We have addressed each of the reviewers' requirements as outlined below, and these are indicated in the manuscript in yellow.

Thank you once again for these valuable suggestions, which have greatly helped us to further improve the quality and clarity of this manuscript.

Answers to academic editor

1. Recommendations to the Author: Abstract_Introduction.

Academic editor's comment: As there is no direct analytical proof such as regression analysis, significance (p-value), statement like this in line 92- Body dissatisfaction significantly affects adolescents' mental health, leading to issues with body weight control, low self-esteem, and poor school performance. This statement could be softened by rephrasing such as “Body dissatisfaction appears to affect/ can be affect”, in order to avoid overgeneralization.

Authors’ response: Thank you for your thoughtful suggestion regarding the statement in line 90. We agree that the original wording may have conveyed a level of certainty that is not fully supported by direct analytical proof in our manuscript. In response to your valuable feedback, we have revised the sentence to: ‘Body dissatisfaction can affect adolescents' mental health, leading to issues with body weight control, low self-esteem, and poor school performance.’

2. Recommendations to the Author: Abstract_Methods.

Academic editor's comment: Specifying the number and demographics of participants upfront like “A total of 24 interviews with adolescents, parents, and teachers in Spain were conducted for this study” would be clearer for the readers.

Authors’ response: Thank you for your thoughtful suggestion. We are specifying the number and demographics of participants upfront in lines 92-93.

3. Recommendations to the Author: Abstract_Conclusion.

Academic editor's comment: Suggest adding implication for broader public health policies to enhance the abstract’s impact.

Authors’ response: Thank you for your thoughtful suggestion. We have added: “Strengthening facilitators and reducing barriers should guide future public health policies" in lines 104-105.

4. Recommendations to the Author: Indroduction_Indicate the study gap.

Academic editor's comment: Clearly articulate the gap in prior studies to increase impact of this study. As an example, statement like this can be used “Existing studies focus primarily on individual factors but lack insights into community-based interventions targeting SE and BI.".

Authors’ response: Thank you for your valuable suggestion. We agree that clearly articulating the gap in prior studies enhances the impact of this research. In response to your comment, we have added the following statement to the manuscript: “However, most studies on SE and BI have focused on individual and group interventions, many of which address pathological situations, but lack perspectives on community-based interventions aimed at tackling these issues” This addition is supported by relevant bibliographic references to strengthen the rationale for our study (lines 159-161).

5. Recommendations to the Author: Indroduction_Structure.

Academic editor's comment: Concluding the introduction with a specific research question or hypothesis would be helpful to set expectations for the readers.

Authors’ response: Thank you for your valuable suggestion. We agree that a clear research question helps to set expectations for the readers, as you mentioned. As our study follows a qualitative research approach, we do not start with a prior assumption but rather seek to explore the phenomenon in question. Therefore, we have concluded the introduction with a specific research question instead of a hypothesis to align with the nature of qualitative inquiry (lines 164-166).

6. Recommendations to the Author: Methods_Recruitment and participants.

Academic editor's comment: As line 190 “Purposive sampling” used in recruitment, elaborate that and address potential biases in using peer nomination.

Authors’ response: Thank you for your valuable comment. We have elaborated on the use of purposive sampling and addressed the potential biases associated with peer nomination in the Limitations section (lines 665-668). Specifically, we discuss how this approach may have introduced biases, such as the influence of group dynamics or social relationships, and the steps taken to mitigate these issues.

7. Recommendations to the Author: Methods_ Data analysis.

Academic editor's comment: In line 237- “Inductive and deductive coding were performed to identify the meaning units of the text.” Explain how inter-coder reliability was ensured during thematic analysis. If it is possible provide an overview of the coding process, including an example of how themes were derived from raw data.

Authors’ response: Thank you for your valuable suggestion. We agree that providing a clear explanation of how inter-coder reliability was ensured and offering an overview of the coding process will help clarify our methodology. In response, we have revised and improved the data analysis section to address your concerns (lines 246-267). We hope this revision addresses your comment and enhances the clarity of our methodology.

8. Recommendations to the Author: Results_Theme Presentation.

Academic editor's comment: Introduce a summary table with the main themes, subthemes, and representative quotes for quick reference.

Authors’ response: Thank you for your valuable suggestion. We have introduced a summary table (Table 2, lines 286 and 291-292) that includes the main themes, subthemes, and representative quotes for quick reference.

9. Recommendations to the Author: Results_Quotes.

Academic editor's comment: Shorten participant quotes to highlight only the most impactful insights.

Authors’ response: Thank you for your valuable suggestion. We have shortened the participant quotes to highlight only the most impactful insights.

10. Recommendations to the Author: Discussion_Policy Implications.

Academic editor's comment: Include examples of how findings can inform school-based policies or public health initiatives.

Authors’ response: Thank you for your valuable suggestion. In response, we have included specific examples of how our findings can inform school-based policies and public health initiatives (lines 642-650 and 654-656). We believe these additions strengthen the practical implications of our findings for both policy and practice.

11. Recommendations to the Author: Discussion_Limitations.

Academic editor's comment: Expend on how the rural Catalonian setting may limit generalizability and suggest strategies for urban applications.

Authors’ response: Thank you for your comment. We have expanded on how the rural Catalonian setting may limit the generalisability of the findings and have suggested strategies for urban applications in the Limitations section (lines 674-683). Specifically, we discuss how socio-cultural and economic factors in rural settings differ from urban areas and propose that future research explore similar interventions in urban schools to better address the challenges and opportunities presented by more diverse and heterogeneous populations.

12. Recommendations to the Author: Conclusion_Recommendation.

Academic editor's comment: Conclusion could’ve been end with a stronger call to action for researchers, educators, and policymakers.

Authors’ response: Thank you for your comment. In response to your suggestion, we have revised the conclusion to more closely reflect the primary findings of the research and provide clearer implications for practice (lines 686-695). The updated conclusion now includes specific recommendations for stakeholders, such as educators, parents, and community leaders, based on the identified barriers and facilitators. These recommendations aim to strengthen the impact of future interventions on self-esteem and body image among adolescents.

13. Recommendations to the Author: Conclusion_Abbreviations.

Academic editor's comment: In line 139 abbreviation like PBI was used but full meaning of the abbreviation was not used prior of that line. Ensure consistency, use full form once then use abbreviation thereafter.

Authors’ response: Thank you for your helpful observation. We apologize for the oversight. As suggested, we have ensured consistency by using the full form of "Positive Body Image" prior to the abbreviation "PBI" in lines 141-142. From that point onward, the abbreviation will be used consistently.

Answers to reviewer 1

1. Comments to the authors_Rigor

Reviewer comment 1: The manuscript presents a qualitative study that carefully examines barriers and facilitators in improving self-esteem (SE) and body image (BI) among adolescents. It appears well-structured, with a clear presentation of themes derived from participant interviews. The qualitative methodology is appropriate for exploring personal experiences and perspectives. However, the description of the data collection methods could benefit from more detail. For example, elaborating on how participants were selected and how interviews were conducted would enhance the rigor by providing clearer context for the findings.

Authors’ response: Thank you very much for your positive feedback and thoughtful suggestions. We appreciate your observation regarding the description of the data collection methods. However, we believe that the level of detail currently provided aligns with the journal's word limit and guidelines for qualitative studies. While we understand the value of elaborating further, we have prioritized maintaining a concise yet comprehensive overview of the methods to ensure clarity without exceeding the recommended length of the manuscript.

2. Comments to the authors_Coherence

Reviewer comment 1: The manuscript is coherent, with logical flow and organization. The themes derived from adolescents, parents, and teachers are clearly presented, highlighting the different perspectives on SE and BI interventions. The discussion effectively ties the qualitative findings back to the existing literature, enabling readers to understand the relevance and implications of the results. Nevertheless, some sections, especially the proposals for future interventions, could be more directly linked back to the barriers identified earlier in the document.

Authors’ response: Thank you very much for your positive feedback and constructive suggestions. We appreciate your observation regarding the need to more directly link the proposals for future interventions to the barriers identified earlier in the manuscript. In response, we would like to clarify that the authors are currently working on developing future intervention proposals in a separate study, where the barriers identified in this manuscript will be directly addressed. This ongoing work aims to build upon the findings from the present study and provide more targeted, practical solutions.

3. Comments to the authors_Scientific Integrity

Reviewer comment 1: The manuscript maintains a good level of scientific integrity. Participants seem to have been treated with respect, and the concerns about ethics are briefly acknowledged, e.g., regarding informed consent. However, providing more explicit details on ethical approval processes, participant anonymity, and potential biases in the research design would contribute to the transparency of the research. It may also be helpful to reflect on potential researcher biases in data interpretation.

Authors’ response: Thank you for your valuable feedback. We appreciate your suggestion to provide more explicit details on the ethical approval processes, participant anonymity, and potential biases in the research design. However, we believe that the ethical considerations section includes the appropriate information in accordance with qualitative research recommendations and the manuscript's length limitations. This study was approved by the Clinical Research Ethics Committee, and the code will be unblinded once the manuscript is accepted (lines 276-277).

To further address the potential for researcher biases in data interpretation, we have ensured inter-coder reliability by having all three researchers independently code the interview transcripts. This approach was taken to minimize bias and ensure a rigorous and transparent analysis process.

4. Comments to the authors_Ethical Considerations

Reviewer comment 1: The manuscript acknowledges the sensitive nature of discussing issues like body image and self-esteem. However, ethical considerations warrant more thorough discussion, especially with regard to adolescent participation. More information about the age range of participants, parental consent, and how sensitive topics were handled during interviews would enhance the ethical robustness of the study. A discussion of how the researchers supported participants who may have experienced distress from topic discussions could also be considered.

Authors’ response: Thank you for your thoughtful comment. We appreciate your concern regarding the ethical considerations, particularly with adolescent participation. We have ensured compliance with all ethical standards, including obtaining parental consent for all participants. Regarding the sensitive nature of the topics discussed, we would like to clarify that none of the participants experienced distress during the interviews. The researchers were mindful of the sensitive nature of the discussions and ensured that the participants felt comfortable throughout the process. It is important to highlight that the individuals who conducted the interviews are clinical professionals who regularly use clinical interviews as a standard tool in their work. However, we acknowledge the importance of supporting participants and will reflect on potential support strategies in future studies.

5. Comments to the authors_Data Collection Details

Reviewer comment 1: It would be good to offer more detail about the methods used for data collection, including sampling methods, interview guide structure, and how interviews were analyzed (e.g., thematic analysis approach).

Authors’ response: Thank you for your suggestion. We would like to clarify that the authors attached the semi-structured interview guide in Supplementary File 2. The authors believe that the information provided regarding the sampling and data collection methods conforms to the guidelines for qualitative studies. However, we have expanded the data analysis section to offer a clearer explanation of the process, specifically the thematic analysis approach used to analyze the interview data. We hope these revisions further clarify the methodology employed in the study.

6. Comments to the authors_Linking

Reviewer comment 1: Findings to Interventions: It is imperative to strengthen the reasoning behind proposed interventions by linking them back to specific barriers more explicitly. Discuss how each suggested intervention directly addresses identified barriers.

Authors’ response: We appreciate your suggestion to strengthen the reasoning behind the proposed interventions by more explicitly linking them to the specific barriers identified in the manuscript. In response, we would like to clarify that the authors are currently working on a separate study focused on developing intervention proposals that directly address the barriers highlighted in this manuscript. This ongoing research aims to ensure that each proposed intervention is closely aligned with the identified challenges, providing more targeted and practical solutions. We look forward to building upon the findings of this study to inform future interventions.

7. Comments to the authors_Diversity of the Sample

Reviewer

---

## [Editor Report · Decision Letter 1]

27 Jan 2025

Key factors in supporting adolescents to achieve high self-esteem and a positive body image: A qualitative community-based study

PONE-D-24-42336R1

Dear Dr. Martinez,

We’re pleased to inform you that your manuscript has been judged scientifically suitable for publication and will be formally accepted for publication once it meets all outstanding technical requirements.

Kind regards,

Mukhtiar Baig, Ph.D.

Academic Editor

PLOS ONE

---

## [Editor Report · Acceptance letter]

PONE-D-24-42336R1

PLOS ONE

Dear Dr. García Martínez,

I'm pleased to inform you that your manuscript has been deemed suitable for publication in PLOS ONE. Congratulations! Your manuscript is now being handed over to our production team.

Kind regards,

on behalf of

Professor Mukhtiar Baig

Academic Editor

PLOS ONE